# Endosperm Persistence in *Arabidopsis* Results in Seed Coat Fractures and Loss of Seed Longevity

**DOI:** 10.3390/plants12142726

**Published:** 2023-07-22

**Authors:** Joan Renard, Gaetano Bissoli, María Dolores Planes, José Gadea, Miguel Ángel Naranjo, Ramón Serrano, Gwyneth Ingram, Eduardo Bueso

**Affiliations:** 1Instituto de Biología Molecular y Celular de Plantas, Universitat Politècnica de València-Consejo Superior de Investigaciones Científicas, Camino de Vera, 46022 Valencia, Spain; joareme@ibmcp.upv.es (J.R.); gaetano.bissoli@mail.com (G.B.); doplanes@gmail.com (M.D.P.); jgadeav@ibmcp.upv.es (J.G.); mnaranjo@ibmcp.upv.es (M.Á.N.); rserrano@ibmcp.upv.es (R.S.); 2Laboratoire Reproduction et Développement des Plantes, ENS de Lyon, CNRS, INRAE, UCBL, F-69342 Lyon, France; gwyneth.ingram@ens-lyon.fr

**Keywords:** seed longevity, seed coat, endosperm, fissures, AtHB25, ICE1, lignin, *Arabidopsis*

## Abstract

Seeds are specialized plant organs that carry, nurture, and protect plant offspring. Developmental coordination between the three genetically distinct seed tissues (the embryo, endosperm, and seed coat) is crucial for seed viability. In this study, we explore the relationship between the TFs AtHB25 and ICE1. Previous results identified *ICE1* as a target gene of AtHB25. In seeds, a lack of ICE1 (*ice1-2*) suppresses the enhanced seed longevity and impermeability of the overexpressing mutant *athb25-1D*, but surprisingly, seed coat lipid polyester deposition is not affected, as shown by the double-mutant *athb25-1D ice1-2* seeds. *zou-4*, another mutant lacking the transcriptional program for proper endosperm maturation and for which the endosperm persists, also presents a high sensitivity to seed aging. Analysis of *gso1*, *gso2*, and *tws1-4* mutants revealed that a loss of embryo cuticle integrity does not underlie the seed-aging sensitivity of *ice1-2* and *zou-4*. However, scanning electron microscopy revealed the presence of multiple fractures in the seed coats of the *ice1* and *zou* mutants. Thus, this study highlights the importance of both seed coat composition and integrity in ensuring longevity and demonstrates that these parameters depend on multiple factors.

## 1. Introduction

Orthodox seeds are structures that can maintain the life of the corresponding future plant in a quiescent state through controlled desiccation, reducing embryo metabolism, and limiting ROS production and damage [1]. Although exceptions exist, e.g., resurrection plants, most plant tissues cannot cope with extreme dehydration. In non-persistent endosperm species, the endosperm is absorbed by cotyledons during seed development and may be absent in mature seeds. In *Arabidopsis thaliana*, it is present as a single layer of live cells surrounding the mature embryo. In some *Arabidopsis* mutants, the endosperm is not absorbed and persists in the mature seed. During seed development, the embryo and the endosperm are primed to cope with dehydration through the accumulation of protective molecules and through specific cellular alterations that protect their cellular and molecular structures [2].

By contrast, the seed coat, a maternally derived, enveloping seed tissue, undergoes a complex developmental program that transforms different cell layers into specific structures that will contribute to a protective isolation barrier (Appendix A). The last step in this process is cell death [3]. The main function of the seed coat is to protect the embryo from mechanical and environmental stresses, but it also has important functions in maintaining seed dormancy and seed dispersal [4,5]. Recently, we demonstrated the importance of lipid polyester barriers within the seed coat, the suberin layer, and the endosperm associated-cuticle, which isolate the embryo from the external environment, limiting air and humidity penetration and ensuring embryonic viability over time [6,7].

Another molecular barrier in the seed coat is formed through the accumulation of proanthocyanidins (PAs), which are crucial to limiting cuticle permeability [8]. These compounds are progressively oxidized during the dry storage of seeds, leading to the browning of the seeds [9]. As a result, an intense dark-brown color of a seed is often used as an indicator of reduced viability. PAs also contribute to the control of seed dormancy [4].

ICE1 was first described as a transcription factor that induces the expression of CBF genes (the DREB1 subfamily) and controls many developmental and adaptation processes, including stomatal formation [10] and cold resistance [11]. It is expressed widely in plants, including in the three different seed tissues [12]. It is also critical in the regulation of endosperm breakdown, through the formation of heterodimers with ZHOUPI (ZOU) This protein dimer is produced since their bHLH domains interact. As ZOU expression is tightly restricted to the developing endosperm, it provides the tissue specificity required for the regulation of the genetic program of endosperm breakdown. [13,14]. The collapse of the abnormal, persistent endosperm in *ice1* and *zou* mutant seeds leads to seed shriveling during desiccation. In addition, embryonic cuticle permeability, an additional apoplastic barrier, is compromised in *ice1* and *zou* mutants since their activity is required for the correct functionality of the embryonic cuticle integrity pathway, which involves the GASSHO1 (GSO1) and GSO2 receptors and the TWISTED SEED1 (TWS1) peptide [15,16]. *ice1* and *zou* mutant seeds also show increased seed primary dormancy [17].

Previous results have indicated that AtHB25, a regulator of seed longevity in several species [7], is a TF expressed in the seed coat during seed development and regulates permeability through lipid polyester accumulation in the cuticle and palisade layer. This mechanism involves the direct regulation of long-chain acyl-CoA synthetase 2 (LACS2), an enzyme that is essential for catalyzing the cutin and so, lacs2 mutant presents higher loss of germination over time because of the seed cuticle incorrect formation. In addition, AtHB25 binds to the *ICE1* promoter region [7]; therefore, in light of the important role of ICE1 in seed development, we explored the potential impact of this regulation. The mutant *athb25-1D*, which ectopically overexpresses *AtHB25*, presents an increase in gibberellic acid and abscisic acid levels in plants and seeds [18,19]. We found that ICE1 is necessary for the low permeability of the *athb25-1D* seed coat; however, this bHLH transcriptional activator does not affect seed lipid polyester accumulation. Rather, the impaired endosperm degradation of *ice1-2* mutants leads to the formation of seed coat fissures. We propose that these fissures compromise the integrity and isolating properties of the seed coat, resulting in a strongly reduced level of seed longevity.

## 2. Results

### 2.1. AtHB25 Regulates ICE1 Expression via Direct Promoter Binding

ChIP-seq analysis of the seed longevity regulator AtHB25 uncovered an AtHB25 binding site at the promoter region of *ICE1* [7]. As shown in Figure 1a, the AtHB25 binding region was found close (−40 nt) to the transcription start site (5’UTR) in the *ICE1* promoter. A motif analysis of the AtHB25 binding region revealed the presence of four potential AtHB25 binding motifs. These were found to be in two distinct domains. In each of these domains, two motifs overlap, one on the forward and the other on reverse strand, such that they share the TAATTA motif (colamp-ATHB25-DAP-Seq), as illustrated in Figure 1b. AtHB25 Motif overlapping occurs at least once in 45% of AtHB25 ChIP seq peaks [7]. This may suggest that the formation of AtHB25 dimers is needed for specific expression regulation. Consistent with this, AtHB25, like other homeobox proteins, has been shown to form homo- and heterodimers [20]. We confirmed the ability of AtHB25 to form homodimers using the Split-Trp technique (Figure 1c), which is a useful method for analyzing transcription factors since protein interaction does not occur in the nucleus, thus avoiding autoactivation [21].

To test that AtHB25 regulates *ICE1* expression, we analyzed the expression of *ICE1* in the *athb25-1D* mutant and in the double mutant *athb22 athb25* [18] in developing seeds from 10 days after pollination (DAP) to 14 DAP between the bent cotyledon stage and mature seed green stage, which coincided with the observed seed coat expression of *AtHB25* [7]. The expression of *ICE1* was increased in *athb25-1D*, and it was not significantly decreased in the double mutant during seed development (Figure 1d). In addition, using ATTED-II (https://atted.jp/, accessed on 1 February 2023), we observed (as shown in Appendix A) a good correlation in terms of gene expression pattern (coex z = 2.67) between *ICE1* and *AtHB25*. These results support the ability of AtHB25 to regulate *ICE1* positively. The fact that *ICE1* was not downregulated in the double mutant in the developing seeds suggests that *ICE1* expression during seed formation is not exclusively dependent on AtHB25, which is probably due to the fact that *ICE1* is expressed in all three seed tissues, i.e., the embryo, endosperm, and seed coat [12].

### 2.2. ICE1 Function Is Crucial for Seed Longevity

AtHB25 is a trans-species seed longevity regulator, and since *ICE1* is a direct AtHB25 target, we investigated the possible role of ICE1 in the process of seed germination loss over time. We grew all plants used to assess seed longevity under the same conditions and at the same time. Thus, we grew AtHB25 gain- and loss-of-function mutants with *ice1-2* and the newly isolated homozygous *athb25-1D ice1-2* double mutant. Visually, the fresh seeds of *ice1-2* present a darker color compared with the wild-type seeds (Figure 2a). We hypothesized that this dark pigmentation, which resembles that of long-term-stored seeds, could be a consequence of the premature oxidation of PAs and be responsible for the higher seed dormancy seen in *ice1-2* seeds [17]. It has previously been shown that *ice1-2* primary dormancy can be suppressed by exogenous gibberellic acid application and after proper seed ripening [17]. We confirmed these results under our study conditions. In addition, the increased dormancy of the *ice1-2* seeds was partially reduced in the double mutant *athb25-1D ice1-2* (Appendix A). This effect could be explained by a previously reported increased level of GA accumulation in *athb25-1D* [18]. To determine whether ICE1 plays a role in seed longevity, we performed a controlled deterioration treatment (CDT), which confirmed the reduced aging resistance of the *ice1-2* seeds (Figure 2b). As a control in this experiment, we included the *athb25-1D* mutant, which presents increased aging resistance (compared to the wild-type) for 80% of its germinated seeds after treatment (for comparison, the wild-type presents 60% germination). We confirmed that *ice1-2* is very susceptible to CDT as its germination was less than half of that of the sensitive mutants *athb22 athb25* and *lacs2-3* [7]. Intriguingly, the germination of *ice1-2* after CDT was not rescued by *athb25-1D* (*athb25-1D ice1-2*) (Figure 2b).

One of the main causes of poor seed viability has been shown to be high seed coat permeability. This parameter is usually measured via formazan formation after seed treatment with tetrazolium salts. *athb22 athb25* presented higher seed coat permeability, which was likely because its seeds accumulate fewer apoplastic lipid polyesters such as cutin and suberin [7] (Figure 2c,d). Most of the *ice1-2* seeds turned red after 48 h of tetrazolium incubation, revealing high seed coat permeability despite having a tiny embryo and suggesting defects in the apoplastic barrier that forms in the seed coat and/or around the embryo, which appears to be equally defective in *ice1-2 athb25-1D* (Figure 2c). 

One plausible explanation for the higher seed coat permeability, primary dormancy, and PA oxidation of the *ice1-2* seeds is that ICE1 could play a role in seed coat apoplastic barrier biosynthesis. The positive regulation of *ICE1* by AtHB25 might also support this hypothesis since AtHB25 regulates cutin and suberin accumulation in the seeds of several species [7]. We conducted Sudan Red staining on delipidated seeds; surprisingly, *ice1-2* seeds showed no reduction in seed coat lipid polyesters compared with the wild-type seeds (Figure 2d). In addition, the increased levels of lipid polyester deposition in the *athb25-1D* seeds were not restored to the WT levels by introducing the *ice1-2* allele, suggesting that increased *ICE1* transcription is not the cause of increased lipid polyester deposition or increased seed longevity in *athb25-1D.*

### 2.3. Endosperm Elimination Is Critical for Seed Longevity

ICE1 is involved in endosperm breakdown during seed maturation through heterodimerization with another bHLH TF, namely, ZOU. To determine whether the seed-aging sensitivity of *ice1* could be related to endosperm persistence, we analyzed the *zou-4* mutant. This mutant, like *ice1-2,* shows impaired embryonic growth and endosperm elimination [14]. Importantly, however, *ZOU* expression is tightly restricted to the developing endosperm [13]. To confirm the artificial aging results obtained for *ice1-2*, we stored the seeds in dry conditions (20–25 °C; 40–60% RH). After one year of storage, the germination of *ice1-2* was strongly reduced compared to the wild type, and only 35% of the seeds showed root extrusion (Figure 3a). We found that the *zou-4* mutant seeds also showed a similar reduction in seed viability after storage. Stratified (to interrupt dormancy) fresh seeds of both mutants were used as controls, and it was found that they showed almost 100% germination (Figure 3a). ICE1 and ZOU form heterodimers to regulate endosperm breakdown. When either protein is absent, seeds present a persistent endosperm and impaired embryo growth. In addition, defects in embryonic cuticle integrity are observed in both mutants [14]. Although the possibility has not been investigated in previous studies, we considered that embryonic cuticle defects could be a factor regulating embryonic viability and could thus explain why *ice1-2* and *zou-4* show reduced seed longevity. To test this hypothesis, we performed CDT on mutants with disrupted embryonic cuticles but lacking a persistent endosperm. In this experiment, we included the double mutant *gso1 gso2* and the *tws1-4* mutant [15,16]. As observed during natural aging, the germination of *ice1-2* and *zou-4* after accelerated seed aging was strongly reduced (lower than 10%) compared with the wild type (higher than 60%). The germination rates of *gso1 gso2* and *tws1-4* were found to be around 50% (Figure 3b). This result suggests that the presence of embryo cuticle defects has only a small impact on seed longevity and that the poor seed viability of *ice1-2* and *zou-4* cannot be explained by this factor. In addition, although the intense seed browning and increased permeability to tetrazolium salts observed in the *ice1-2* seeds were also observed in the *zou-4* seeds, these phenotypes were not observed in either the *gso1 gso2* or *tws1-4* seeds (Figure 3c,d). These results suggest that the reduced seed longevity in the *ice1-2* and zou-*4* seeds was a consequence of the loss of endosperm degradation and that embryonic cuticle defects contribute very little to seed impermeability defects or decreased longevity. Furthermore, we acquired histological sections of the developing seeds (12 DAP). After Toluidine Blue staining (Figure 2e), we found that, as expected, *ice1-2* presented impaired endosperm degradation compared with the wild type. The double mutant *ice1-2 athb25-1D* seeds resembled the ice*1-2* seeds. Accordingly, the *ice1-2 athb25-1D* seedlings showed a highly permeable cuticle (Appendix A), while the *athb25-1D* seedlings resembled the wild-type seeds. From these results, we conclude that AtHB25 is not directly involved in the maintenance of embryonic cuticle integrity, and it cannot compensate for the defects in the endosperm elimination of *ice1-2* mutants.

### 2.4. AtHB25 and ICE1 Regulate Polyphenolic Content in Seeds

Both suberin and lignin are synthesized in the Arabidopsis seed coat where they function as barriers between the embryo and the environment. Suberin presents polyaliphatic domains (polyester) and polyphenolic domains [24]. The lignin polymer results from the oxidative coupling of monolignols and coniferyl, sinapyl, and p-coumaryl alcohols [25]. Both compounds are thought to be important for the preservation of seed longevity and impermeability [6,26]. Unfortunately, there are no existing methods allowing for the measurement of suberin polyphenolics and lignin separately. To gain an indication of the total polyphenolic content of the seeds, we utilized the acetyl bromide method since it is the technique with the best reported recovery [27]. We found that *athb25-1D* seeds showed the highest acetyl bromide soluble lignin percentage (% of ABSL) after spectrophotometric quantification. Consistent with this finding, the *athb22 athb25* double mutant seeds showed lower polyphenolic content than the wild type (Figure 4). Levels similar to those found in the *athb22 athb25* double mutant seeds were found in the *zou-4* and *ice 1-2* seeds (Figure 4).

It is important to note that the alterations in polyphenolic levels in *zou-4* and *ice 1-2* could have been induced by morphological differences (such as reduced embryo size); thus, the decreased polyphenolic content in *zou-4* and *ice 1-2* could have origins different from those ascribed to *athb22 athb25*. Given that the intensity of Sudan Red staining (which detects suberin and cutin but not lignin) was not markedly different in the *ice 1-2* mutant compared to the wild type, it is possible that the observed polyphenolic reduction in *zou-4* and *ice 1-2* was due to a reduction in lignin or a lignin-like biopolymer (Figure 2). This notion is also supported by the fact that although the double mutant *athb25-1D ice1-2* shows similarly increased levels of suberin (Sudan Red) in comparison to those of the single *athb25-1D* mutant, its total polyphenolic content was reduced (intermediate between WT and ice1-2 levels). Thus, we suggest that ZOU/ICE1 heterodimers in the endosperm are required for the production of a polyphenolic biopolymer other than suberin and that might contribute to seed longevity and seed coat impermeability. The tissue in which this polymer resides and its true function remain unresolved.

### 2.5. Endosperm Persistence Leads to Fissures in the Seed Coat

In prior studies, the seeds of the *ice1-2*, *athb25-1D ice1-2*, and *zou-4* mutants presented seed shriveling [14,28]. These previous studies suggested that shriveling predominantly occurs in the testa overlying the abnormally persistent and subsequently dehydrated endosperm. We observed mature seeds using Scanning Electron Microscopy (SEM) and detected the presence of fissures in the seed coats of the *ice1-2* and *zou-4* mutant seeds but not in any of the observed WT seeds (Figure 5). These fissures were detected in most of the observed *ice1-2* and *zou-4* mutant seeds. Fissures mostly occurred at epidermal cell borders (Figure 5), but large breaks and other kinds of fissures were also observed (Appendix A). In many cases, the epidermal cells were deformed and shriveled. Interestingly, the fissures were mainly located in seed coat areas overlying the endosperm (Appendix A). These fissures in the seed coat could cause reduced seed longevity, seed coat permeability, and the premature oxidation of PAs in the *ice1-2* and *zou-4* seeds since their presence would nullify the protection provided by the isolating seed coat. In developing seeds, these seed coat fissures were visualized during seed drying but not before (Appendix A), suggesting that they are the result of seed coat collapse due to the volume loss of the dehydrating non-eliminated endosperm.

## 3. Discussion

Seeds are specialized organs whose development requires tightly coordinated differentiation in all three genetically distinct tissues. The endosperm is a nourishing tissue that sustains the embryo, and it has been shown to play a role in seed dormancy since its absence has a positive effect on physical dormancy in calcareous grassland plants [29], although its role in seed longevity is not so clear. Several analyses have found that species with non-endospermic seeds present resistance to seed deterioration when compared with endospermic seeds [30,31]. In rice, the endosperm is unable to accumulate protective compounds such as antioxidants or repair proteins such as PIMT [32], possibly increasing ROS accumulation around the embryo. Herein, we report that endosperm degradation is essential for seed coat integrity and thus for the proper protection of the offspring. The ICE1/ZOU complex coordinates this process, and in its absence, the endosperm persists, thereby hindering embryo growth [14]. Although the growing embryo combined with the persistent endosperm together exert an increased internal pressure in *zou* mutants [33] (which may contribute to seed coat fractures), visible fissures appear only in the last stages of maturation and at the start of desiccation (Appendix A). The simplest explanation for this is that the immature endosperm in dicots is more than 90% water; thus, the loss of this water during seed desiccation causes cracks in the seed coat (Appendix A). Moreover, we noticed this volume loss is recovered when the seeds were rehydrated (Figure 2c,d and Figure 3), indicating that the volume of persistent endosperm cells varies strongly with water potential, which is contrary to the case of embryonic cells. The dramatic deformation caused by the volume loss during seed dehydration likely breaks apart cell walls that may also have been weakened by increased tension at earlier developmental stages. The resulting fractures compromise seed coat isolation, and the resulting influx of oxygen would rapidly damage the embryo. One visible effect of the penetration of oxygen in the seeds is the precocious seed browning observed in the *ice1* and *zou* mutants caused by the rapid oxidation of PAs. In addition, like seeds with abnormal integument differentiation or seed coat lipid polyester deposition, the *ice1* and *zou* mutant seeds presented increased permeability to tetrazolium salts [4,34], thus confirming that in these seeds the isolation of the embryo has been defective. We have noticed that premature seed browning is also present in aborted seeds [35], suggesting the formation of seed coat fissures via similar processes.

Our results using mutants with reportedly normal seed coats but strong defects in the embryo cuticle, such as *gso1 gso2* and *tws1-4*, confirm that the tetrazolium salt test primarily measures seed coat permeability, but not embryo cuticle permeability, in fresh seeds. The embryo cuticle remains permeable to different compounds before germination [36], and this likely includes tetrazolium salts. It is important to note that our results suggest that embryo cuticle integrity may play a minor, but nonetheless significant role, in ensuring seed longevity. Plants with defective embryonic cuticles encounter difficulties in seedling establishment when they are sown in soil. However, when these mutants are germinated in vitro, they do not present germination defects but still require long periods of culturing in high humidity to allow for acclimation. Under normal growth conditions (in the wild), the embryo cuticle is likely essential for seed germination, including in the process of embryo release from the seed coat [15].

In previous studies, we found that AtHB25 is a trans-species regulator whose ectopic overexpression extends seed life span in *Arabidopsis*, tomato, and wheat. The mechanism triggered by this TF was determined using ChIP seq analyses performed during seed formation. AtHB25 directly activates the transcription of the gene encoding LACS2 (long-chain acyl-CoA synthetase 2), a lipid polyester biosynthetic protein that catalyzes the acyl-CoA intermediates, thereby activating the polymerization process [37], and NHO1, a glycerol kinase. The consequence of the increase in *LACS2* expression is the overaccumulation of 18:2 α,ω-dicarboxilic acid but also of ferulate through unknown mechanisms. Finally, this gene expression change appears to cause the thickening of suberin and cuticle layers, resulting in the better insulation of the embryo from the external environment [7].

ChIP seq analysis identified 146 AtHB25 binding sites during seed maturation. Curiously, motif overlapping was present in 45% of the AtHB25 ChIP seq peaks [7] (Figure 1b), suggesting that protein dimerization may be important for DNA binding and transcription regulation. The most enriched genomic region was found in the 5´promoter region of *ICE1*, and RTqPCR analysis revealed positive regulation of *ICE1* by AtHB25 (Figure 1d). Until now, ICE1 and ZOU have only been described with respect to their role in seed dormancy but not in longevity. The loss of function of these TFs deepens primary dormancy and increases ABA levels. However, it appears that the aberrant endosperm of loss-of-function mutants contributes to the dormancy phenotype [17]. It is plausible that the compromised isolation of seeds and an enhanced oxidation of seed components including PAs affect seed dormancy through an endosperm-dependent mechanism. Since *ICE1* is one of the main target genes of the trans-species regulator of seed longevity AtHB25, it was essential for us to study its role in seed longevity. Our results show that *ice1-2* seeds are very sensitive to aging both in an otherwise wild-type background but also in combination with *athb25-1D*, which normally improves longevity. Consistent with this, although the reduced longevity of the *ice1* mutants is likely due to high seed permeability, *ice1-2* seeds are not impaired in terms of lipid polyester production.

The seed coat is critical for the maintenance of embryonic viability and consists of four to five layers of dead cells in which different polymers are accumulated to ensure a low permeability. In *Arabidopsis*, suberin is accumulated in the palisade layer (a subepidermal layer). The deposition of tannins takes place in the brown pigment layer and in the inner integument, including the cell wall in the endosperm-associated cuticle [38]. The accumulation of suberin in the testa is rather rare, and *Arabidopsis* is thus unusual in this respect. However, cutin is present in the seed cuticle layer in most plant species. Finally, a role played by lignin, a polymer of monolignol involved in maintaining seed coat impermeability in *Arabidopsis*, has been suggested through the study of the function of TT10 laccase (Pourcel et al., «*TRANSPARENT* TESTA10 Encodes a Laccase-Like Enzyme Involved in Oxidative Polymerization of Flavonoids in *Arabidopsis* Seed Coat».) [39] but also using the overexpression of the COG1 transcription factor, which positively regulates seed coat peroxidases [6]. Bissoli et al. [40] also showed a positive correlation between the amount of lignin and seed viability in different pepper species. Our results show that ice*1-2* and *zou-4* seeds contain lower amounts of polyphenolic compounds (likely lignin or lignin-like) than wild-type seeds. Such a reduction, if it occurs in the testa, could render this tissue more fragile and contribute to the appearance of cracks because of endosperm persistence. However, since ZOU only functions in the endosperm, the underlying mechanism remains enigmatic, and it could be related to other effects such as alterations in the different *ice1* and *zou* mutants. Surprisingly, and undocumented in the published literature until now, AtHB25 positively regulates seed polyphenolic content since gain-of-function mutants present higher levels; in addition, this regulation could be dependent upon ICE1 since the double mutant *athb25-1D ice1-2* exhibits lower levels than wild-type seeds. Determining the spatial distribution of polyphenolics in each mutant background will be an important next step in understanding the relationships between these phenomena.

Our findings demonstrated that correct endosperm maturation is essential to preserving the low permeability of seed coats in *Arabidopsis*, i.e., the main barrier that protects embryo viability. Moreover, we have confirmed that ICE1 is a AtHB25 target, and the lack of this bHLH TF or ZOU, another TF necessary for endosperm breakdown, generates fissures in the seed coat and impedes regular lignin accumulation in seed.

In conclusion, this study demonstrates that multiple physical changes in the seed coat, and within internal seed compartments, can contribute to seed longevity, thus underlining the complexity of attempting to understand this trait without the aid of in-depth structural and compositional data.

## 4. Materials and Methods

### 4.1. Plant Material and Growth Conditions

For this study, we used the following Arabidopsis thaliana mutant lines: gain-of-function *athb25-1D* [18] and loss-of-function *athb22 ahtb25* (*SALK_017963*; SALK_014023) [18], *ice1-2* (SALK_003155) [10], *lacs2-3* (GABI_368C02) [41], *zou-4* (GABI_584D09) [13], *gso1 gso2* (SALK_064029; SALK_130637) [42], and *tws1-4* [16]. The double mutant *athb25-1D ice1-2* was developed in this study. All lines are in the Col-0 genetic background, and the Col-0 ecotype was used as the WT reference. All plants of each experiment were grown simultaneously in growth chambers under long day conditions at 22 °C and 70% RH. All T-DNA mutans are loss-of-function mutants, and the positions of T-DNA insertion can be checked in http://signal.salk.edu/cgi-bin/tdnaexpress. Gene Annotation: AtHB25 (At5g65410); AtHB22 (At2g36610); ICE1 (At3g26744); ZOU (At1g49770); GSO1 (At4g20140); GSO2 (At5g44700); TWS1 (At5g01075); and LACS2 (At1g49430).

### 4.2. Seed-Aging Treatments, Seed Dormancy Test, and Seed Germination

In this study, seeds were aged by storing them for one year at room temperature (for seed dormancy, they were stored for one and two weeks). The artificial aging treatment used was the Controlled Deterioration Treatment or CDT, which consisted of aging seeds for 14 days in an enclosed receptacle containing a saturated salt solution at 37 °C. Then, seeds were ethanol- and bleach-sterilized and sown in solid MS media containing sugar. Seeds were stratified for 3 days at 4 °C, except in seed dormancy experiments. Germination ratios were assessed after 10 days to ensure all plants had germinated, including *ice1-2* and *zou-4* seeds, which have small embryos with retarded growth. We verified that after 10 days, the germination ratios did not increase in any experiment.

### 4.3. Seed Coat Analyses and Polyphenolic Analysis

Seed coat permeability test and seed coat lipid polyester staining were carried out as previously described [7,18]. The polyphenolic content was assessed as described by Moreira-Vilar et al. (2014) [27], and the percentage of ABSL was calculated according to method reported by Rains et al. (2018) [43].

### 4.4. Scanning Electron Microscopy

Dried seeds were gold-coated and then surface-scanned with the ZEISS Ultra-55 Scanning Electron Microscope operating at the lowest voltage. Developing seeds were rapidly and directly observed using a HIROX 3000 at −20 °C to minimize water loss. Image acquisition was also performed at the minimum voltage.

### 4.5. Gene Expression Analyses

Total RNA was extracted from developing seeds according to the protocol reported by Oñate-Sánchez and Vicente-Carbajosa (2008) [44], and it was then purified and DNase-treated using E.Z.N.A. Plant RNA Kit (Omega Bio-tek, Norcross, Georgia). A total of 3 μg of RNA was reverse-transcribed using the Maxima first-strand cDNA synthesis kit (ThermoFisher Scientific, Waltham, MA, USA). RTqPCR analysis was performed using QuantStudio 3 (Thermo Fisher Scientific) and the 5x PyroTaq EvaGreen qPCR MixPlus (ROX; Cultek S.L.U., Madrid, Spain). The reference gene used was *PP2AA3* [45]. Relative mRNA abundance was calculated using the comparative ΔCt method [46]. Primers utilized are as follows: *ICE1-qRT-F GGCCAGCAAGCTAGAGTTGA*, *ICE1-qRT-R ATGGTAGCGAGCAACAGACC*, *PP2A3-qRT-F ACCTGCGGTAATAACTCATCTA*, and *PP2A3-qRT-R: CCGAACATCAACATCTGGGTC*.

### 4.6. Toluidine Blue Staining

Etiolated seedlings were stained to assess cuticle integrity using Toluidine Blue Solution (0.05% Toluidine Blue, 0.1% Tween20) for 5 min. Later, the stain was removed using consecutive water washes. Excess water was drained, and plants were imaged using Leica binoculars.

Developing seeds (12 DAP) were fixed, embedded, and cut in accordance with the method reported by Creff et al. (2019) [15]. Sections were stained with the same Toluidine Blue Solution for 5 min, washed, placed in water, and imaged using a Zeiss AxioImager microscope.

### 4.7. Split-Trp

AtHB25 coding sequence was cloned in the *pCTrp* and *pNTrp plasmids* [47] processes using the following primers: AtHB25-common-F CTGGCCATTACGGCCATGGAGTTTGAAGACAACAAC, AtHB25-pCTrp-R TTAGGCCGAGGCGGCCAGTGGTTGGTCTTGTTCATGATG, and AtHB25-pNTrp-R GGCCGAGGCGGCCTCATGGTTGGTCTTGTTCATG. Empty plasmids were used as negative controls. The yeast strain used was CRY1 Mata (-A, -H, -L, -U, -W). Two hybrid yeasts were grown in SD media lacking uracil and leucine for the control growth assay and lacking uracil, leucine, and tryptophan and supplemented with 100 µM of CuSO_4_ for the interaction assay.

## Figures and Tables

**Figure 1 plants-12-02726-f001:**
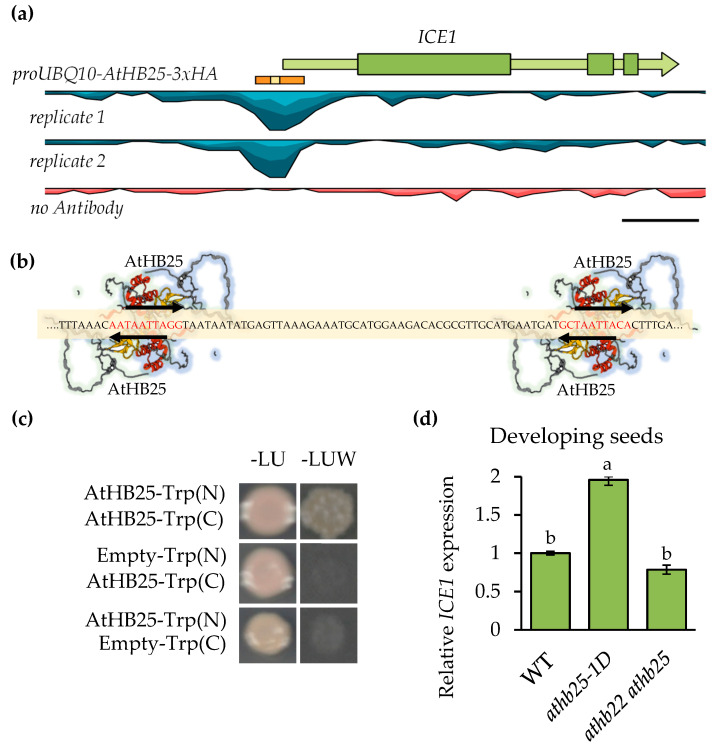
AtHB25 positively regulates *ICE1* expression and forms homodimers. (**a**) Schematic representation of the sequence enriched in the AtHB25 ChIP seq close to the *ICE1* gene, which is illustrated with an orange bar, and its genomic context. Within this orange area, the magnified area (**b**), including the binding motifs of AtHB25, is marked in yellow. The sequence alignment of the two ChIP replicas is shown in blue, and the no-antibody replicate for this genome region in red. Scale bar: 1 Kb. (**b**) Region inside the ChIP seq peak that contains four binding motifs for AtHB25 depicted by arrows and red letters. They overlap on the forward and reverse strands, sharing the TAATTA sequence. The transcription factor AtHB25 is represented by the model obtained from AlphaFold [22,23]. The Zinc-Finger domain is colored in red, and the Homeobox domain is colored in yellow. Proteins binding in the forward motif are highlighted in light blue, while those binding in the reverse motif are highlighted in light green. (**c**) Split-Trp result confirming the formation of AtHB25 homodimers. (**d**) *ICE1* expression analysis in AtHB25 mutants in developing seeds ranging from 10 days after pollination (DAP) to 14 DAP. Expression values are relative to the housekeeping gene *PP2A3* and are normalized to WT. Results consist of the average of three biological replicates. The error bars denote standard errors. One-way ANOVA and a subsequent Tukey’s test were performed to identify the statistically different groups (*p* < 0.05).

**Figure 2 plants-12-02726-f002:**
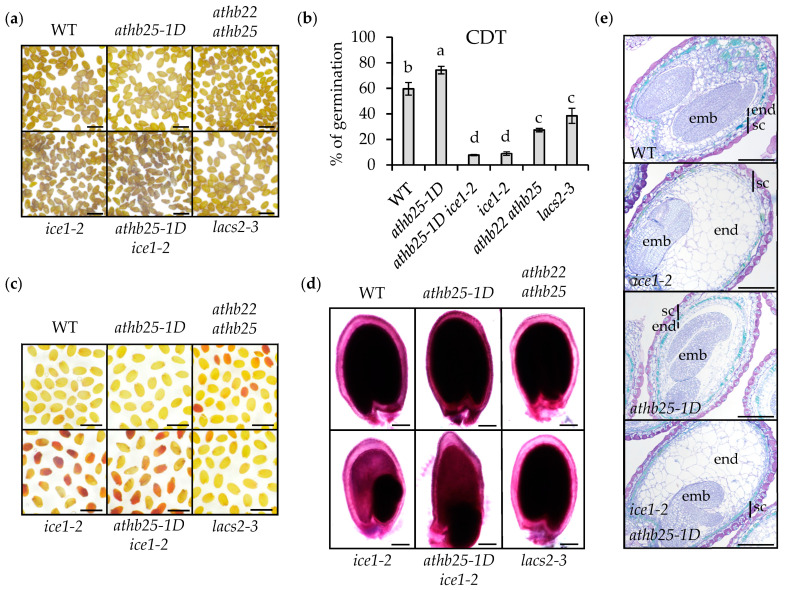
ICE1 is crucial for seed longevity and seed coat impermeability. (**a**) Representative images of dry seeds imaged 3 weeks after seed ripening. Scale bars: 1 mm. (**b**) Percentage of germination after a Controlled Deterioration Treatment (CDT, 14 days at 37 °C and 75% RH). Results are the average germination scores of more than 100 seeds from three replicates obtained from different plants grown at the same time and under the same conditions. Error bars denote standard errors. One-way ANOVA, followed by Tukey’s test, was performed to identify the statistically different groups (*p* < 0.05). (**c**) Seed coat permeability test using tetrazolium salts for 48 h. Images are representative of three different biological replicates. Scale bars: 1 mm. (**d**) Representative images of Sudan Red staining applied to delipidated seeds for seed coat lipid polyester barrier visualization. Scale bars: 100 µm. (**e**) Toluidine blue staining of LR white sections of 12-days-after-pollination seeds. In addition, the endosperm layer is still not a one-cell layer as the seed is still developing. emb: embryo, end: endosperm, and sc: seed coat. Scale bars: 100 µm.

**Figure 3 plants-12-02726-f003:**
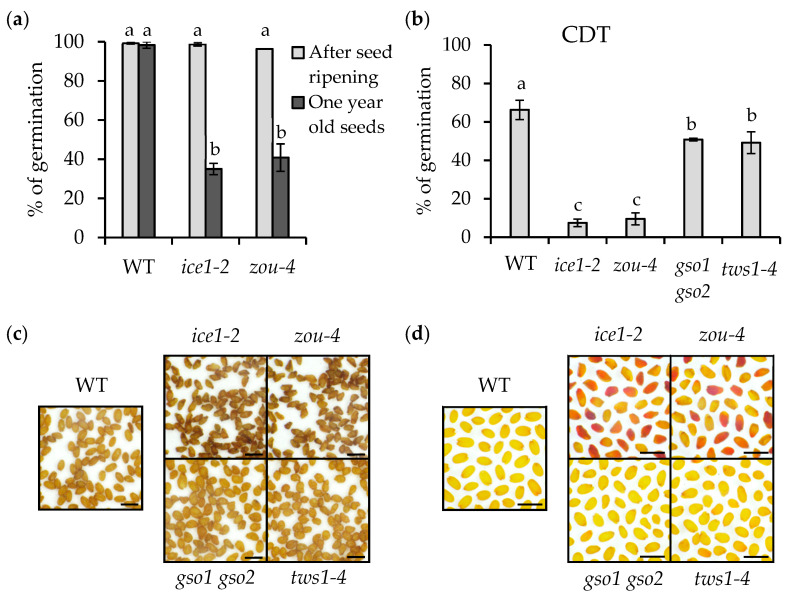
Endosperm elimination is critical for seed longevity and permeability. (**a**) Percentage of stratified seeds germinated after ripening and after one year of dry storage. (**b**) Percentage of germinated seeds after a Controlled Deterioration Treatment (CDT, 14 days at 37 °C and 75% RH). Results of (**a**,**b**) are the average germination scores of more than 100 seeds from three different seed replicates obtained from different plants grown simultaneously under the same conditions. Error bars denote standard errors. One-way ANOVA and a subsequent Tukey’s test were performed under each condition to identify the statistically different groups (*p* < 0.05). (**c**) Representative images of dry seeds imaged 3 weeks after seed ripening. (**d**) Seed coat permeability tests using tetrazolium salts and conducted for 48 h. Images of (**c**,**d**) are representative of three different biological replicates. Scale bars: 1 mm.

**Figure 4 plants-12-02726-f004:**
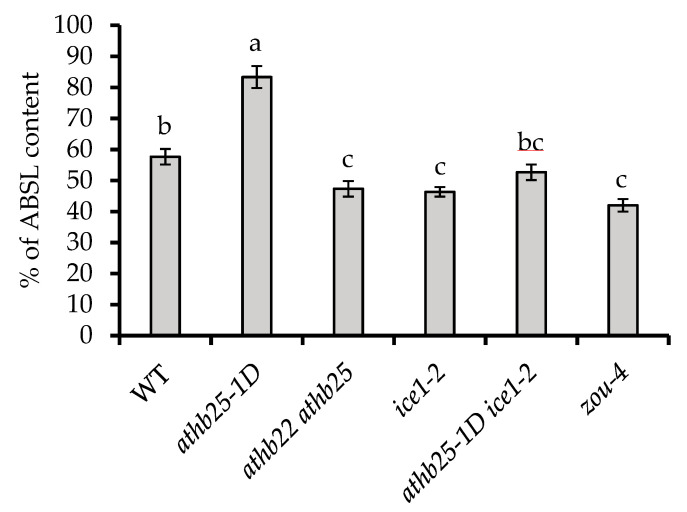
AtHB25 and ICE1 influence seed polyphenolic content. Total acetyl bromide soluble polyphenolic extraction from seeds. The results are the average of 5 samples. %ABSL: acetyl bromide soluble lignin percentage. Error bars denote standard errors. One-way ANOVA and a subsequent Tukey’s test were performed to identify the statistically different groups (*p* < 0.05).

**Figure 5 plants-12-02726-f005:**
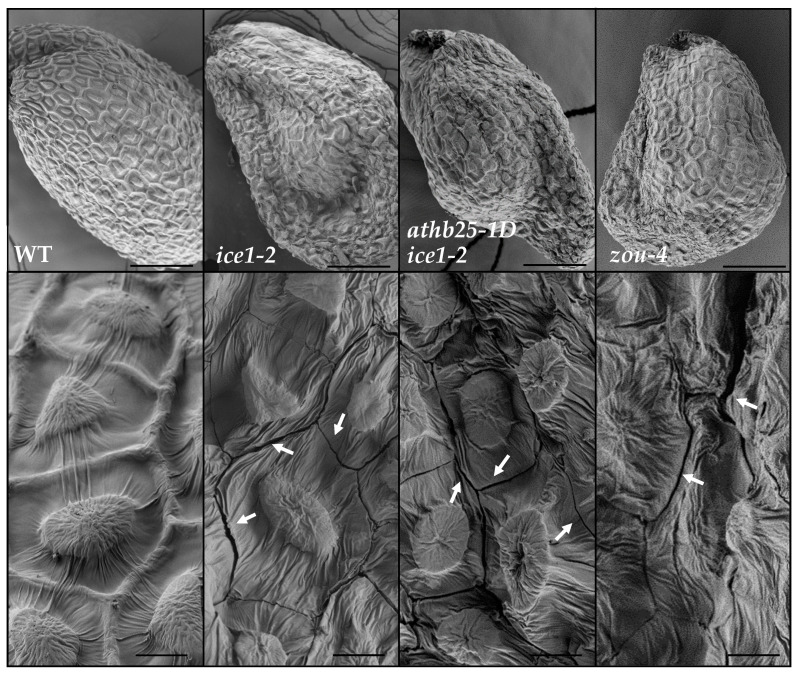
Seed fissures visualized in *ice1-2*, *athb25-1D ice1-2*, and *zou-4* seed coats via scanning electron microscopy. Images are representative of each genotype, and fissures were observed in almost all analyzed mutant seeds but not in WT seeds. Fissures tend to be located in areas that do not overlie the compressed embryo but overlie the dehydrated non-eliminated endosperm (Appendix A). The most common fissure pattern was located at epidermal cell junctions, but other fissure patterns were also observed (Appendix A). (**Up**): Whole seed. (**Down**): Magnification. Arrows point to the presence of fissures. Scale bars: Up: 100 µm; down: 10 µm.

## Data Availability

All data generated are included in the manuscript, including the main text and the supplemental data.

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
