# Peer review of "Endosperm Persistence in Arabidopsis Results in Seed Coat Fractures and Loss of Seed Longevity"

_plants, 2023, doi:10.3390/plants12142726_

Round 1
Reviewer 1 Report
This is the revision of the paper submitted by Renard et al. to Plants.
In this paper, the authors investigate the function of AtHB25 and ICE1 during seed viability and longevity. To perform the research the authors applied several methods and used several single and double mutants.
The data create a clear story. However, to improve the quality of this manuscript, I have several comments and questions.
General comments
Please describe better the mutant lines used in this research. What kind of mutation do they have, stop codon, in which part of the gene is T-DNA insertion? It would be good to show the gene structure and place of T-DNA insertion. This information can be added at the beginning of the Results.
When you are mentioning binding motifs, it would be good to show also protein structures.
In which seed tissue, embryo, endosperm, seed coats AtHB25 and ICE1 are expressed?
Detailed comments
Abstract: Lines 16-19. These two sentences are unclear. It should be mentioned that additional mutants were used as control.
Introduction: Line 38. Instead of ‘Recent studies in our group, demonstrated…” should be “Recently, we demonstrated…).
Lines 48-55. What is the relation between genes mention in this part? Do they create some complex?
Line 60. Please avoid using shortcuts. It should be “… increased gibberellic and abscisic acid levels”. Especially, that in line 120 you have the full name “gibberellic acid”
Lines 60-61. The sentence “We found that….” The second part of the sentence does not sound scientific.
Results. Line 84. 10-14 DAP – which seed developmental stage it is? qPCR experiment should be performed using dissected seed compartments: embryo, endosperm, and seed coats.
Discussion. Line 281. “Several meta-analyses…”. What does it mean? Metabolomic-analyses?
Methods. Section ‘Accession numbers’ is missing. Please mention the gene numbers.
Toluidine blue staining: Seeds – Did you remove the wax before staining? How thick were the slides during cutting on the microtome?
Figure 1(d). It would be better to use separated (dissected tissues) and seeds collected at the same DAP. How many seeds per sample did you use? Statistics – should be performed ANOVA and post hoc test.
Figure 2 (a, c, d) Please add scale. (b) Statistics - each column should have a letter above. Dashes above columns 3 and 4 and 5 and 6 suggest that you performed e.g. comparison between these two samples using a t-test. The highest value should get the letter “a” –not WT but athb25-1D
Figure 3 (a-b). Statistics. For (a) should be performed ANOVA + post hoc test individually for “after ripening’ and “one year old seeds”. (b) The same as in Fig, 2b.
Figure 4. Statistics. The same as in Fig, 2b.
Author Response
This is the revision of the paper submitted by Renard et al. to Plants. In this paper, the authors investigate the function of AtHB25 and ICE1 during seed viability and longevity. To perform the research the authors applied several methods and used several single and double mutants. The data create a clear story. However, to improve the quality of this manuscript, I have several comments and questions.
Thank you for your thorough revision. Now the manuscript will be of better quality.
General comments
Please describe better the mutant lines used in this research. What kind of mutation do they have, stop codon, in which part of the gene is T-DNA insertion? It would be good to show the gene structure and place of T-DNA insertion. This information can be added at the beginning of the Results.
All mutants, except for tws1-4 are T-DNA mutants. T-DNAs were in exons in all SALK mutants, and in introns in GABI mutants as you can check in http://signal.salk.edu/cgi-bin/tdnaexpress. However, all T-DNA mutants were loss of function in the gene where it was inserted. athb25-1D is an activation tagging mutant whose T-DNA is located 1000 pb from 3´. The increase of ATHB25 transcript produced by this insertion it can be checked in reference 17. tws1-4 is a CRISPR mutant as described in reference 15 where the is produced an insertion of 1 cytosine.
This information was described in subsection 4.1 “Plant material and growth conditions”. The information about mutants is used to be in Material and Method and we really think that addition of this information in “Results” could be repetitive for lectors. The author agrees with the reviewer that the information about the consequence of the mutation should be appear in the text and accordingly we added in lines 381 and 382 the words loss of function and gain of function. It also was included at the end of this subsection the sentence “All T-DNA mutans are loss of function mutants and positions of the T-DNA insertion can be checked in http://signal.salk.edu/cgi-bin/tdnaexpress to the lectors that want to check T-DNA position.
When you are mentioning binding motifs, it would be good to show also protein structures.
The figure has been modified to show DNA bindings domain of the transcription factors.
In which seed tissue, embryo, endosperm, seed coats AtHB25 and ICE1 are expressed?
ICE1 is expressed in endosperm, seed coat and in the embryo. (It was indicated in results just before figure 1, new line 102). However, it was also included in the introduction.
It is the ZOU expression which makes the tissue specificity in endosperm breakdown program, as ZOU expression is restricted to the endosperm [13,14]. This was indicated in the discussion, now is also in the introduction.
AtHB25 is expressed in seed coat formation, but also in the embryo as it is described in reference 7. However, the expression in the seed coat is the one responsible of the increased seed longevity and lipid polyester disposition as these traits present maternal heritance in these athb25 mutants.
As both, ICE1 and AtHB25 were expressed in the seed coat, it was logic to try to understand the role of ICE1 in the maternal AtHB25-seed longevity.
We include the sentence “Previous results indicated that AtHB25, TF expressed in seed coat development (line 60)
Detailed comments
Abstract: Lines 16-19. These two sentences are unclear. It should be mentioned that additional mutants were used as control.
The author agrees and this information was included in the abstract.
Introduction: Line 38. Instead of ‘Recent studies in our group, demonstrated…” should be “Recently, we demonstrated…).
The author agrees and this information was included.
Lines 48-55. What is the relation between genes mention in this part? Do they create some complex?
ZOU and ICE1 are co-expressed in the endosperm and interact via their bHLH domains. Heterodimerization of ZOU and ICE1 is necessary for their binding to specific targets. This information was included.
Line 60. Please avoid using shortcuts. It should be “… increased gibberellic and abscisic acid levels”. Especially, that in line 120 you have the full name “gibberellic acid”
The author agrees. This was corrected.
Lines 60-61. The sentence “We found that….” The second part of the sentence does not sound scientific.
The sentence was changed for “we found that ICE1 is necessary for the low permeability of athb25-1D seed coat, however this bHLH transcriptional activator does not affect seed lipid polyester accumulation”.
Results. Line 84. 10-14 DAP – which seed developmental stage it is? qPCR experiment should be performed using dissected seed compartments: embryo, endosperm, and seed coats.
The sentence “between bent cotyledon and mature seed green stage” is included in the text. Since we have short time to reply, this experiment is impossible to perform but we will take into consideration in future. Dissection of seed coat compartments is not possible for us. We do not have such technology for the precise dissection and low sample RNA analysis. Anyway in the future we are planning to do single cell sequencing in seed.
Discussion. Line 281. “Several meta-analyses…”. What does it mean? Metabolomic-analyses?
Since this term is confused, we have changed for “several analyses”.
Methods. Section ‘Accession numbers’ is missing. Please mention the gene numbers.
Gene annotation data was included in methods.
Toluidine blue staining: Seeds – Did you remove the wax before staining? How thick were the slides during cutting on the microtome?
Wax was not removed. The thickness of the slides was 1 micrometer.
Figure 1(d). It would be better to use separated (dissected tissues) and seeds collected at the same DAP. How many seeds per sample did you use? Statistics – should be performed ANOVA and post hoc test.
Tissue dissection is a difficult experiment, but we will take in consideration in next approach. At least we use 500 developing seed to get enough RNA to perform CDNA synthesis. Anova analysis have been performed leading same statistical differences as before with t-test
Figure 2 (a, c, d) Please add scale. (b) Statistics - each column should have a letter above. Dashes above columns 3 and 4 and 5 and 6 suggest that you performed e.g. comparison between these two samples using a t-test. The highest value should get the letter “a” –not WT but athb25-1D
Done
Figure 3 (a-b). Statistics. For (a) should be performed ANOVA + post hoc test individually for “after ripening’ and “one year old seeds”. (b) The same as in Fig, 2b.
Done
Figure 4. Statistics. The same as in Fig, 2b.
Done
Reviewer 2 Report
The authors showed that the importance of both seed coat composition and integrity in ensuring longevity in Arabidopsis by the analysis of ice1, athb25, zou, gso1gso2, and tws1-4 mutants. The minor suggestions are as below:
1. It will be clearer for readers to add the “Arabidopsis” in title.
2. The authors concluded that AtHB25 positively regulates ICE1 expression. I think that more experiments such as LUC assay should be conducted and added in Figure 1 to confirm it.
3. the % followed the number should be deleted in Y axis in Figure 2b, 3b, and 4.
4. Line 229, the word “differences” repeated.
5. Line 389, what is mean of “RT” (root temperature)?
Author Response
Thank you for your revision. Now the manuscript will be of better quality.
The authors showed that the importance of both seed coat composition and integrity in ensuring longevity in Arabidopsis by the analysis of ice1, athb25, zou, gso1gso2, and tws1-4 mutants. The minor suggestions are as below:
- It will be clearer for readers to add the “Arabidopsis” in title.
Arabidopsis was included in the title.
- The authors concluded that AtHB25 positively regulates ICE1 expression. I think that more experiments such as LUC assay should be conducted and added in Figure 1 to confirm it.
Thanks for the suggestion we will perform experiment in the future. Anyway, in previous study the direct interaction was proved using ChIP-qPCR.
- the % followed the number should be deleted in Y axis in Figure 2b, 3b, and 4.
Done
- Line 229, the word “differences” repeated.
Deleted
- Line 389, what is mean of “RT” (root temperature)?
we changed in manuscript.
Reviewer 3 Report
The article is well written and interesting in terms of seed physiology and survival. The article is suitable for publication but with a few changes:
It would be worthwhile for the authors to mention the cuticle which is formed by the cells of the embryo and is an additional apoplastic barrier.
Since the authors mention apoplstic barriers in seeds, it would be worth mentioning symplastic barriers between the embryo and tissues of maternal origin. Are there any known plasmodesmata between the endosperm and the embryo?
The authors could give a diagram of the seed and indicate the tissues they are discussing and the barriers between them.
Please correct the species name Arabidopsis using italics (references)
Please add Fig S3 to main body of manuscript. On the photos please add markers of of seed coat, endosperm layer and embryo.
- markers of seed coat, endosperm, embryo.
Author Response
The article is well written and interesting in terms of seed physiology and survival. The article is suitable for publication but with a few changes:
Thank you very much for your revision.
It would be worthwhile for the authors to mention the cuticle which is formed by the cells of the embryo and is an additional apoplastic barrier.
This information is included in the introduction (line 54).
Since the authors mention apoplstic barriers in seeds, it would be worth mentioning symplastic barriers between the embryo and tissues of maternal origin. Are there any known plasmodesmata between the endosperm and the embryo?
The author apologizes but this topic is out of his knowledge. However, as far we now, the embryo and the endosperm are simplastically isolated, as well as the endosperm and the seed coat.
The authors could give a diagram of the seed and indicate the tissues they are discussing and the barriers between them.
We have drawn a diagram of the seed showing the different tissues and barrier we can fin in Arabidopsis seed (Figure S1, line 36)
Please correct the species name Arabidopsis using italics (references)
Done
Please add Fig S3 to main body of manuscript. On the photos please add markers of of seed coat, endosperm layer and embryo.
Figure S3 now in included as Figure 2e. Endosperm is not a layer yet as seed are still developing, however differences on endosperm degradation can be appreciates. This information as well that cut are not completely in the middle of the seed in WT and athb25-1D, has been included in the figure legend.
Reviewer 4 Report
The authors made interesting findings about ICE1 and seed longevity. The authors present clearly results and discuss the possible explanations. The main issue I have with this manuscript is that it does not provide enough background knowledge for me to understand all the hypotheses the authors made.
Line 27, 'though' should be 'through'?
Line 28, what exceptions?
Line 47, could you give more background about CBF? It is sudden to mention CBF. If I remembered correctly, CBF is also a TF to induce cold stress genes so it is a little bit confusing for the whole sentence whether you want to talk about CBF or ICE1.
Line 61, please explain more about what AtHB25 does for the seed coat permeability. When you said ICE1 does not do this, what do you mean?
Line 112, could you introduce what is seed aging? I am confused about why the authors wanted to study ICE1 in seed aging without any knowledge about seeds.
Line 373, the conclusion is vague, could you highlight your findings and summarize your perspectives of your study? The authors discussed a lot in the previous paragraphs, but I think a summary is needed.
For the gene expression method, what primers did you use to detect your genes of interest?
Some typos need to be fixed.
Author Response
The authors made interesting findings about ICE1 and seed longevity. The authors present clearly results and discuss the possible explanations. The main issue I have with this manuscript is that it does not provide enough background knowledge for me to understand all the hypotheses the authors made.
The author agrees with all suggestions, and we really appreciate your work in the revision.
Line 27, 'though' should be 'through'?
Done
Line 28, what exceptions?
Resurrection plants, it is included in manuscript in the new version.
Line 47, could you give more background about CBF? It is sudden to mention CBF. If I remembered correctly, CBF is also a TF to induce cold stress genes, so it is a little bit confusing for the whole sentence whether you want to talk about CBF or ICE1.
ICE1 (inducer of CBF expression1) is a TF that induces the expression of DREB1 subfamily genes (CBF, DRE/CRT-binding proteins). So, they are related but not the same family of TF. This information is included in the manuscript.
Line 61, please explain more about what AtHB25 does for the seed coat permeability. When you said ICE1 does not do this, what do you mean?
The author agrees and the required information about AtHB25 is included.
Line 112, could you introduce what is seed aging? I am confused about why the authors wanted to study ICE1 in seed aging without any knowledge about seeds.
We have changed the concept of seed aging for an explanation of what it consists of. Seed aging is the loss of germination over time.
Line 373, the conclusion is vague, could you highlight your findings and summarize your perspectives of your study? The authors discussed a lot in the previous paragraphs, but I think a summary is needed.
Thanks for your suggestion. We have added at the end of discussion a summary with the main findings of the study.
For the gene expression method, what primers did you use to detect your genes of interest?
Thank you for the observation, we listed in the section 4.5 Gene expression analyses the 4 primers used.
Reviewer 5 Report
I like data in this paper, but I think there is a flaw that the authors have not addressed. I believe the simplest explanation for their data is that fractures in the seed coat form during seed desiccation when the embryo does not grow to fill and occupy the volume of the embryo sac. Because the endosperm in dicots is greater than 90% water, loss of this water during seed desiccation causes cracks in the seed coat. This does not occur in wild type controls because this volume is occupied with seed storage components (starch, protein bodies, lipids, and structural carbohydrates). Sunken depressions can be observed in ice2 seeds, but not in controls.
Figure 2d shows tetrazolium staining of seeds. In each of these images, TZ penetrates the seeds well. Poor penetration would be the absence of or considerably weaker staining of the embryo. I believe the difference in staining is the size of the embryo. Embryos normally stain darker with this assay. Please address this criticism.
I would like to see an analysis of these phenotype in backcrossed ice2 mutants. Endosperm phenotypes are inherited and segregate differently from seed coat mutants. By backcrossing the mutants, the timing of the mutation can be determined. Are these phenotypes deriving from an endosperm effect or seed coat effect? I think this needs to be done to clarify which of these structures causes the loss of seed longevity. Doing this also addresses the issue I rose in the first paragraph above.
Minor issues that need to be addressed:
ICE1 binding site is at -40 nt (not 40 nt after) from tc. start site.
Fig. 1 should use the term replicate, instead of replica.
Need to explain the phenotype of the lacs2 mutant when first mentioned, not in the discussion.
The text is well-written, however, there number of times where the sentence structure is awkward and the reader needs to re-read the sentence to understand the authors intent.
Author Response
I like data in this paper, but I think there is a flaw that the authors have not addressed. I believe the simplest explanation for their data is that fractures in the seed coat form during seed desiccation when the embryo does not grow to fill and occupy the volume of the embryo sac. Because the endosperm in dicots is greater than 90% water, loss of this water during seed desiccation causes cracks in the seed coat. This does not occur in wild type controls because this volume is occupied with seed storage components (starch, protein bodies, lipids, and structural carbohydrates). Sunken depressions can be observed in ice2 seeds, but not in controls.
Thanks for the comment but in fact we have the same hypothesis about the formation of cracks (lines 294-296). “Upon desiccation, those persistent endosperm cells lose volume making the seed coat collapse internally in areas overlying the endosperm (Figure S6). We noticed this volume loss is recovered when seeds are rehydrated (Figure 2c, d and Figure 3). But we included your explanation since is clearer.
Figure 2d shows tetrazolium staining of seeds. In each of these images, TZ penetrates the seeds well. Poor penetration would be the absence of or considerably weaker staining of the embryo. I believe the difference in staining is the size of the embryo. Embryos normally stain darker with this assay. Please address this criticism.
This a very good observation, that reinforce our results. ice1 mutant and double mutant seeds are more stained despite they have a tiny embryo. A sentence was added in line 151.
I would like to see an analysis of these phenotype in backcrossed ice2 mutants. Endosperm phenotypes are inherited and segregate differently from seed coat mutants. By backcrossing the mutants, the timing of the mutation can be determined. Are these phenotypes deriving from an endosperm effect or seed coat effect? I think this needs to be done to clarify which of these structures causes the loss of seed longevity. Doing this also addresses the issue I rose in the first paragraph above.
Crosses have been performed with ice1-2 as a mother plant in our group for this paper and for other studies. The phenotype was completely restored to wild type in offspring (easyliy to see as seedlings were much bigger than ice1-2), suggesting that indeed the heritance is not maternal. However, we do not have any supporting pictures. Another supporting fact for this is that zou-4 also present same phenotype as ice1-2 and its expression is completely restricted to the endosperm. Endosperm and seed coat are simplistically isolated, so the migration of a TF is no likely probable. This supports the idea that indeed, is the problem with the endosperm degradation the one responsible of seed coat fractures and diminished seed longevity.
So indeed, they are derived for an endosperm-embryo effect, as the endosperm is not degraded and embryo cannot occupy the main seed cavity.
Minor issues that need to be addressed:
ICE1 binding site is at -40 nt (not 40 nt after) from tc. start site.
Thanks for the observation, now it is corrected in the text.
Fig. 1 should use the term replicate, instead of replica.
Done
Need to explain the phenotype of the lacs2 mutant when first mentioned, not in the discussion.
We have added a sentence in the introduction explaining the lasc2 phenotype (line 61-64).
Round 2
Reviewer 1 Report
The authors improved the paper as it was recommended. I don't have any additional comments.
Author Response
Thank you very much for your revision.
Reviewer 4 Report
Thank you for the revision, the authors solve my confusion.
Author Response
Thank you very much for your revision.
Reviewer 5 Report
The authors answers to my previous concerns on heritability of the phenotype and TZ staining are satisfactory.
Minor usage changes needed that will be picked up by the copy editors.
Author Response
Thank you very much for your revision.